# Comparative Analysis of Docosahexaenoic Acid (DHA) Content in Mother’s Milk of Term and Preterm Mothers

**DOI:** 10.3390/nu14214595

**Published:** 2022-11-01

**Authors:** Giulia Vizzari, Daniela Morniroli, Francesca Alessandretti, Vittoria Galli, Lorenzo Colombo, Stefano Turolo, Marie-Louise Syren, Nicola Pesenti, Carlo Agostoni, Fabio Mosca, Maria Lorella Giannì

**Affiliations:** 1Neonatal Intensive Care Unit, Fondazione IRCCS Ca’ Granda-Ospedale Maggiore Policlinico, 20122 Milan, Italy; 2Department of Clinical Sciences and Community Health, University of Milan, 20122 Milan, Italy; 3Dialysis and Transplant Unit, Fondazione IRCCS Ca’ Granda Ospedale Maggiore Policlinico, Pediatric Nephrology, 20122 Milan, Italy; 4Fondazione IRCCS Ca’ Granda Ospedale Maggiore Policlinico, Pediatric Area, 20122 Milan, Italy

**Keywords:** DHA, fatty acid, mother milk, newborn, preterm infants

## Abstract

Objectives and Study: Docosahexaenoic acid (DHA) plays an essential role in infants’ development. Maternal diet and breastmilk are the primary DHA sources for newborns. This single-center observational study aimed to compare the DHA content in mother’s milk of preterm mothers with that of term ones, and to investigate the changes in mother’s milk DHA content according to the week of the gestational age. Methods: A food frequency questionnaire (FFQ) was submitted to each mother to estimate the DHA intake during the last trimester of pregnancy, and the mother’s milk was collected between 24 and 96 h post-partum. Results: Women who gave birth prematurely showed a lower content of mother’s milk DHA than the term ones (0.51; IQR 0.38–0.6% FA vs. 0.71; IQR 0.52–0.95% FA; *p* = 0.001). In the multivariate linear regression analyses, for each additional week of gestational age, there was an increase in DHA content in the mother’s milk (0.046% FA; CI 95% 0.018–0.074; *p* < 0.001). Conclusions: Our results suggest that breast milk may not be sufficient to fully satisfy the recommended DHA intake in preterm infants. This study may represent a starting point to investigate new possible DHA supplementation strategies, especially for the late and moderate preterm infants.

## 1. Introduction

Preterm babies are infants born before 37 weeks of gestational age (GA). This population is classified according to gestational age as extremely (less than 28 weeks), very (28 to 32 weeks), moderate (32 to 34 weeks), and late preterm (34 to 37 weeks) babies [1]. Almost 15 million babies, more than one out of ten, are born prematurely each year, and about 80% of them are moderate and late preterm babies [1,2]. Prematurity represents a crucial public health problem worldwide, and its complications are the leading cause of mortality and morbidity in children at up to 5 years of life [1].

Premature birth is related to the underdevelopment of the infant’s tissues and organs and leads to acute complications such as infections, necrotizing enterocolitis (NEC), respiratory distress syndrome (RDS), and intraventricular hemorrhages (IVH), or long-term adverse outcomes such as bronchopulmonary dysplasia (BPD) and motor, neurodevelopmental, and growth impairment, and it often shows crucial nutritional gaps [3]. However, research has focused on new support and care strategies for extremely preterm or very low birth weight (birth weight less than 1500 g) or extremely low birth weight (birth weight less than 1000 g) newborns, leaving out a crucial percentage of premature infants, such as late preterm babies. Failing to recognize the specific medical vulnerabilities of the latter group of preterm infants could lead to a disadvantage regarding short- and long-term health outcomes.

Advances made in neonatology have enabled an improvement in the survival and a reduction in the comorbidities related to prematurity [1]. Nutrition and breast milk are paramount in neonatal care and support [4,5].

An ever-growing body of evidence underlines the importance of breast milk, not only in terms of its nutritional value but as a fundamental link between the mother and her child within a tightly interconnected system [6]. Breast milk encompasses a living and dynamic biological system that allows the exchange of biochemical and immunomodulatory signals from the mother to her baby, thus promoting the newborn’s growth and consequent future health benefits [7]. 

Human milk lipids supply almost 50% of the newborn’s energy intake. In addition, the intake of lipids and particularly their derivate eicosanoids contribute to a wide variety of structural features such as cell membrane composition and several biologic functions including immunomodulation, lipoprotein metabolism, gastrointestinal functions, overall growth, and neurodevelopment. In the last few decades, breast milk-based long-chain polyunsaturated fatty acids (LC-PUFAs), particularly docosahexaenoic acid (DHA), have caught researchers’ interest because of their paramount role in the development of the fetus and the newborn [8]. In fact, DHA is involved in the early implantation and placental process, influencing vascular remodeling and angiogenesis and, through intense storage in fetal structures (adipose tissue, central nervous system, and retina) play a pivotal role in neurogenesis phenomena, synaptic transmission, and the transduction of cellular signals [9,10]. Recently, the latest evidence has further underlined the positive effects of DHA. If on the one hand its early effects on child’s development and visual acuity are undoubted, the possible long-term effects in the adult population remain to be understood [11]. De Jong et al. evaluated the relationship between DHA levels in neonatal blood at birth and neurocognitive development analyzed at 9 years of life. The results revealed that children with lower blood levels of DHA at birth had lesser neurological dysfunction [12]. Consistently, in a double-blind randomized study, Helland et al. evaluated the cognitive outcome at 4 years of the life of children born to mothers supplemented with different doses of DHA from 18 weeks of pregnancy up to 3 months after delivery. Their findings showed that children of women who were supplemented with the highest dose of DHA had an improvement in IQ [13]. Furthermore, Hellström et al. in a recent multicenter randomized clinical trial, enrolled a total of 206 infants born before 28 weeks of gestation. About half of the enrolled infants were enterally supplemented with arachidonic acid and DHA within three days of birth and until 40 weeks of the postmenstrual age; the second group was treated according to the Swedish standard of care. Their results showed a 50% reduction in the risk of developing premature retinopathy (ROP) among the infants who received LC-PUFA supplementation compared to those who received the standard of care [14].

The positive effects of DHA appear to be of particular importance in the preterm population. In fact, the accumulation of LC-PUFAs in the fetus is closely related to the maternal diet. Newborns whose mothers have high levels of dietary LC-PUFAs have higher levels of LC-PUFAs too, not only at birth but also in the first weeks of life [15]. Consequently, during pregnancy and lactation, the maternal DHA requirements increase by 100–200 mg per day compared to the 250 mg indicated as the daily reference intake for the general population [16]. While the transplacental transfer of LC-PUFAs occurs throughout pregnancy, during the third trimester, the placenta facilitates a preferential transfer of DHA over other fatty acids, called “biomagnification”, thus favoring adequate DHA storage in fetal tissues [17]. For this reason, preterm infants often show a DHA deficiency compared to full-term ones [18,19]. Moreover, preterm infants appear to be less capable of synthesizing DHA from precursor fatty acids and are often exposed to an insufficient postnatal intake [19]. On the other hand, the adequate transfer of DHA in the fetal period seems to be crucial for neonatal growth and development [11]. Data from the literature suggested a progressive storage of DHA in the retina and the prefrontal cortex starting from fetal life up to approximately the first two years of life; subsequently, the DHA storage remained substantially stable to guarantee correct neurocognitive and neurobehavioral maturation [11]. Therefore, the first two years of life represent an important window of opportunity for children to ensure an adequate intake of DHA [20].

Despite this growing evidence of the positive effects of DHA in the developing fetus, the infant, and the child, there is still no universal consensus in the literature on the recommended intake during pregnancy and for the infant and breastfeeding woman. In addition, there is also a scarcity of universal recommendations on possible supplementation both in the case of pregnancies at risk of preterm delivery and in the case of breastfeeding a premature baby. This lack of consensus could be due to the wide variability of DHA metabolism, which seems to be influenced by several factors including sex, genetics, endogenous metabolism, diet, and lifestyle [11]. 

In fact, in a double-blind, multicenter, randomized controlled trial, Makrides et al. aimed at evaluating whether an increased supplementation of DHA (via rich fish oil capsules providing 800 mg/day of DHA) during the last trimester of pregnancy could improve the health outcomes of both the mother and child regarding, e.g., a lower incidence of maternal depressive symptoms and the better neurocognitive development of the newborn. At the end of the study, 2327 women were enrolled, but the authors failed to find a statistically significant difference in the health outcomes between the enriched DHA supplementation group and the control one [21].

Human milk represents the primary postnatal source of DHA in breastfed babies. Nevertheless, there is no consensus in the literature regarding the DHA content of breast milk or its value related to prematurity and, therefore, whether breast milk may compensate for the DHA deficit typical of preterm newborns [18,19,22,23,24]. Moreover, while DHA supplementation is currently recommended for those infants born extremely preterm or with very low birth weights [18,25,26], there is a lack of evidence regarding moderate or late preterm birth [27].

Considering the role of LC-PUFAs in neonatal growth and development, we conducted a study to compare the DHA content in preterm mother’s milk with that of term mothers. As a secondary objective, we aimed to investigate the changes in the DHA content in mother’s milk according to the week of gestational age. 

## 2. Materials and Methods

### 2.1. Design and Setting

This observational, prospective, single-center study was conducted at Fondazione IRCCS Ca’ Granda Ospedale Maggiore Policlinico of Milan between August and November 2021. The Ethics Committee of the Fondazione IRCCS Ca’ Granda Ospedale Maggiore Policlinico approved the study (protocol code 791_2021bis and date of approval CE 6 July 2021).

### 2.2. Population

We enrolled 191 women older than 18 years of age who gave birth preterm (<37 weeks of gestation) and at term (≥37 weeks of gestation) at our Center between August and November 2021. We excluded women who were affected by any clinical condition that could interfere with lipids’ metabolism and absorption and women who did not wish to or could not breastfeed. Furthermore, since part of maternal data was obtained through direct interviews or by filling out a written questionnaire (Food Frequency Questionnaire), we decided to exclude women who did not have a good understanding or did not speak the Italian language to avoid bias related to translation. Enrollment took place between 24 and 96 h post-partum. A written, informed consent was obtained by the enrolled mothers.

### 2.3. Data Collection

Maternal medical history was collected through clinical charts, especially for data related to the clinical course of pregnancy such as order of pregnancy, comorbidities, or chronic diseases. We obtained the main anthropometric variables (weight, height, and pre- and post-pregnancy BMI), maternal lifestyle, and socio-demographic information by a direct, structured interview. We collected data regarding educational level (middle/high school or bachelor’s degree), employment (yes or no), smoking habits (we considered “smokers” all women who regularly smoke at least five cigarettes/day) [28], and practicing of regular physical activity (yes or no). 

A validated questionnaire to record adherence to the Mediterranean diet and a food frequency questionnaire (FFQ) was submitted to each of the enrolled mothers to investigate the estimated intake of DHA and EPA and any fatty acid supplementation during the last trimester of pregnancy [29]. Neonatal data were collected from the medical charts and included gestational age, anthropometric parameters at birth (weight, length, and head circumference), infants’ gender, twin or singleton status, mode of delivery, and mode of feeding. 

### 2.4. Samples Collection

Biological samples’ collection was performed between 24 and 96 h post-partum [30,31,32]. Mother’s milk specimens were collected into a sterile tube. The physiological variations of lipids in breast milk were taken into consideration both during the day and during a single breastfeeding session [33]. Every mother’s milk sample (0.3 mL) was obtained through hand expression at the end of a single session of breastfeeding or breast pumping. For each woman, we collected two distinct samples collected in the morning (09:00–11:00) and in the afternoon (16:00–18:00) [30], which were subsequently pooled in a single sterile tube. Considering the circadian variability of breastmilk lipid content, samples’ pooling was performed in order to obtain a sample that was as representative as possible. To evaluate the DHA content in neonatal blood, we obtained a blood sample by pricking the infant’s heel and collected it onto a pretreated filter paper card (Whatman 903 BHT). The blood sample collection was performed during a clinical test, such as blood gas analysis or newborn metabolic-screening test, so as to void performing invasive and painful procedures for the sole purpose of the study. All samples were immediately stored at −20 °C after collection or pooling and then sent to the laboratory for further analysis. All data were anonymously collected in a predefined online database.

### 2.5. Laboratory Analysis

All the samples were analyzed within a maximum of one week from the collection. From each sample, 25 μL of mother’s milk was extracted and transferred into a different vial and then methylated with 700 μL of hydrochloric acid in methanol (Sigma Aldrich, Saint Louis, MO, USA). After being heated at 90 °C for an hour and then cooled at 4 °C for ten minutes, potassium chloride (2 mL) and Hexane (400 μL) were added to the solution. All samples were first vortexed and then centrifuged at 3000 rpm for 10 min. Then, the upper layer was collected and transferred into a gas chromatography vial. The Shimadzu Nexis GC-2030 (Shimadzu, Japan) gas-chromatographer equipped with a 30m fused silica capillary column FAMEWAX Restek (Restek S.R.L., Cernusco sul Naviglio MI, Italy) was used for fatty acid profile evaluation. The gas chromatography results were analyzed using LabSolution software (ver. 5.97 SP1, Shimadzu Corporation, Kyoto, Japan). Single fatty acids were expressed as the relative percentage of total fatty acids.

### 2.6. Statistical Analysis

Statistical analysis was performed with the IBM SPSS program, version 25.0 (SPSS Inc., IBM Company, Chicago, IL, USA). 

Continuous variables were tested for normality using the Kolmogorov–Smirnov test. The variables with a normal distribution were presented as mean ± standard deviation (SD), and the ones with a non-normal distribution were expressed as median and interquartile range (IQR). Categorical variables were expressed as absolute frequencies or percentages. The comparison of continuous variables between the term vs. the preterm group was performed by *t*-test or Mann–Whitney U Test for variables with normal and non-normal distributions, respectively. A comparison of categorical variables was conducted using the Pearson’s Chi-square Test. 

A multivariate linear regression analysis was performed to evaluate the effect of gestational age on DHA content in breast milk and neonatal blood, adjusted for maternal BMI, smoking habit, and maternal DHA intake. 

For the analysis, both preterm and term mothers were further divided into two sub-groups according to the DHA intake during the third trimester of pregnancy being either higher or lower than the one recommended by EFSA (>100–200 mg of DHA/day+ EPA-DHA requirement for the general population, equal to 250 mg/day) [16].

## 3. Results

During the study period, from August to November 2021, 293 women met the inclusion criteria. A total of 60 women denied their consent to the study while 22 were excluded for their lack of understanding the Italian language or because they would not breastfeed. A total of 191 women were divided as follows: 103 mothers that gave birth at term and 88 that had preterm births. The study’s flow chart is shown in Figure 1.

Regarding the neonatal population, we enrolled a total of 121 preterm newborns among which 93 were late preterm (gestational age 34–36 weeks), 23 moderate preterm (gestational age 32–33 weeks), and 5 very preterm (gestational age < 32 weeks). Approximately 55% of the preterm newborns were twins, which explains the discrepancy between the preterm mothers’ population and their infants. Furthermore, most of the newborns showed a birth weight that was adequate for their gestational age. The characteristics of our neonatal population are described in Table 1.

The mean maternal age was 34 ± 5 years. The mean pre-pregnancy BMI was 21.8 ± 3.6 kg/m^2^. Half of the women (54%) were primiparous, which would be slightly increased if we consider only the preterm births cohort (58%). Most women were Caucasian (93%), and a small minority were Hispanic (3%) and Asian (4%). The percentage of sampled women holding a bachelor’s degree was 88.1%, and most were employed (86%). As for smoking habits, a wide fraction of the population (84.5%) stated they never smoked during pregnancy nor in the first days of the postnatal period. 

Regarding the dietary habits in our cohort, the mean DHA intake during the last trimester of pregnancy was 518 ± 218 mg/diet. It was estimated that 71% of the enrolled mothers met the EFSA recommendations regarding the DHA intake during pregnancy. However, the percentage of preterm mothers who did not meet the aforementioned recommendations was significantly higher than that of the term ones (37% vs. 20%, *p* = 0.010) (Figure 2).

A total of 173 mother’s milk samples and 193 neonatal blood samples were collected. Table 2 shows the fatty acid compositions of the mother’s milk samples.

Differences were observed in the mother’s milk fatty acids profiles comparing full-term and preterm mothers. Women who gave birth prematurely showed a lower content of mother’s milk DHA as compared to the term ones (0.51, IQR 0.38–0.6% FA vs. 0.71, IQR 0.52–0.95% FA; *p*= 0.001). On the other hand, no difference among groups was found in the mother’s milk ARA content (1.00; IQR 0.83–1.18% FA vs. 1.05, IQR 0.88–1.27% FA; *p* = 0.110).

Likewise, we found a statistically significant difference in neonatal blood DHA content between term and preterm newborns (4.72, IQR 4.14–5.28% FA vs. 3.44, IQR 2.73–3.85% FA, *p* < 0.001).

In the univariate linear regression analyses, for each additional week of gestational age there was an increase in the DHA content in both mother’s milk (0.044% FA; CI 95% 0.020–0.069 *p* < 0.001) and neonatal blood (0.239% FA; CI 95% 0.151–0.325, *p* < 0.001) (Figure 3 and Figure 4). 

After adjusting for the maternal BMI, smoking habits, and the maternal DHA intake, gestational age remained independently associated with DHA content of both mother’s milk (0.046% FA; CI 95% 0.018–0.074, *p* < 0.001) and neonatal blood (0.237% FA; CI 95% 0.150–0.325, *p* < 0.001). 

We analyzed the relationship between the DHA content in mother’s milk and neonatal blood. In the linear regression analysis, after adjusting for the maternal BMI, smoking habits, and maternal DHA intake in the last trimester of pregnancy, a statistically significant increase in neonatal blood DHA content was correlated with an increasing DHA concentration in mother’s milk (1.29% FA; CI 95% 0.85–1.73, *p* < 0.001.

## 4. Discussion

DHA plays a crucial role in neonatal growth and development. The maternal diet is a major contributor to the storage of DHA in fetal tissues [8]. A possible seasonal influence on fat consumption during pregnancy should be taken into consideration. Van Staveren et al., in a 2-year study of Dutch women, found that winter and spring appeared to be the seasons with the highest fat intake [34]. On the other hand, Watson et al. highlighted a higher dietary fat intake in spring and summer than in winter and autumn [35]. In our study, the FFQ submitted to each mother to investigate the estimated intake of DHA and EPA referred to a period between spring and summer, comprising the reported seasons with a higher fat intake.

Considering either dietary intakes and the specific DHA supplementation, all women showed an overall adequate intake of DHA compared to the EFSA reference values [16].

We can speculate that this finding was the result of a careful information campaign implemented in our center in accordance with national and international recommendations regarding the adequate intake of DHA during pregnancy and beyond [16,36,37]. Juber et al., in their intervention study, evaluated the levels of DHA in breast milk before and after an appropriate educational intervention, through which mothers were informed not only of the importance of DHA for the proper growth and development of their baby but also of how the intake of neonatal DHA was closely related to maternal intake both during pregnancy and lactation. Following the educational intervention, Juber et al. reported a significant increase in both the maternal daily intake of DHA and the DHA levels in breast milk [37].

On the other hand, in our study, the percentage of women who did not reach the recommended EFSA intake was significantly higher among women who gave birth prematurely compared to those who gave birth at full term. Consistently, Middleton et al. highlighted the important contribution of the maternal intake of DHA in reducing the risk of preterm birth [38]. Accordingly, Carlson et al., in a multi-center, double-blind, randomized clinical trial, enrolled 1100 pregnant women between 12 and 20 weeks of gestation [39] and divided them into an experimental group that received 1000 mg DHA/day and a control group with 200 mg/day of DHA supplementation. At the end of the study, the authors concluded that women with a higher intake of DHA had a lower incidence of premature birth (<34 weeks) [39]. These findings could be partly explained by the anti-inflammatory and immunomodulatory activity of DHA, which has already been widely investigated in previous studies [40,41].

A preterm delivery limits the transplacental transfer of DHA and, consequently, the neonatal lipid reserves. At the same time, preterm infants exhibit an enzymatic immaturity that reduces DHA’s endogenous synthesis, limiting its role as an adequate source [19,42,43]. Accordingly, in our study, the DHA content in neonatal blood increased in parallel with gestational age. These findings highlight that even mild prematurity may pose a risk for DHA deficiency compared to term births. 

Breast milk represents the main source of DHA for both term and preterm newborns, but the specific relationship between the maternal diet and the DHA content in breast milk is still uncertain. The storage of LC-PUFAs in human milk mostly represents the mother’s LC-PUFAs dietary intake in the last 48 h [44], with a maximum concentration in human milk between 6 and 12 h after DHA-rich food ingestion [45,46]. Bzikowska-Jura et al. confirmed a direct correlation between the usual intake of omega 3 LC-PUFAs and their breast milk levels [47]. 

In a recent comprehensive study, the MEDIDIET working group investigated the role of the maternal diet on breast milk macronutrients, particularly in its fatty acids content [48]. The authors enrolled 300 Italian mothers who were clinically healthy and exclusively breastfed their infants, with no use of formula milk. This population was divided into five groups based on five different diet models. From their results, it was pointed out that the diet group characterized predominantly by “vitamins, minerals and fibres” and “fatty acids with fins” (fish, seed oil, and white and red meat) showed higher omega-3 fatty acids content in their breast milk, including DHA, than other dietary groups.

According to the literature, one of the factors in our study that could influence the concentration of DHA in breast milk was the gestational age at the time of delivery. Fares et al. evaluated the association between the human milk fatty acid profiles in women who gave birth prematurely and had different maternal characteristics [49]. The study concluded that the arachidonic acid and DHA concentrations in breast milk had a positive correlation with gestational age at delivery, and a negative correlation with maternal age (cut off 34 years) and pre-eclampsia [49].

Similar to Fares et al.’s results, in our study, the concentration in mother’s milk directly correlates with gestational age, with an increase in the DHA content of 0.046% FA for each additional week of pregnancy, equal to 0.5% FA for a 28-week preterm compared to a 38-week term infant. Consistently, other authors have demonstrated that preterm birth is one of the main factors influencing the concentrations of LC-PUFAs in human milk [50,51]. Likewise, a wide variety of other factors, such as dietary, genetic, sociodemographic, health, and environmental features could modulate DHA concentration in human milk [52]. Previous studies provided diverging data on the DHA content of breast milk [53]. Kuipers et al. found that gestational age is inversely related to the content of LC-PUFAs in breast milk, with greater differences in the colostrum than in the mature milk [32]. It appears reasonable that heterogeneity in methodologies (for instance, sample collection, storage, and others) may be at the heart of these discrepancies and may explain the differences among the available studies. On the other hand, there seems to be a unanimous consensus in the literature regarding the variation in the human milk DHA content during lactation. Women who deliver prematurely show a reduction in the DHA levels in their breastmilk throughout lactation [28,29].

Considering that the risk of DHA deficiency is inversely related to gestational age, breast milk alone cannot adequately meet the DHA requirements of preterm infants, including the late preterm ones, whose neural development is also suboptimal and whose nutritional requirements are often approximated to those of full-term infants [54,55]. To date, there are limited studies evaluating the DHA requirements in the different subgroups of preterm births, as the recommendations currently in force—although referring to all preterm babies—have proven to be especially indicative for those born at a birth weight less than 1250 g [56].

## 5. Conclusions

In conclusion, not only very preterm newborns, but also moderate and late preterm infants are at high risk of fatty acid deficiency due to the sudden interruption of DHA transfer through the transplacental route and the consequently lower lipid storage and possible concomitant development of comorbidities related to prematurity. Since the DHA content in both breast milk and neonatal blood progressively increases with increasing gestational age, breast milk alone may not be sufficient to fully satisfy the nutritional recommendations regarding DHA intake in preterm infants.

Our large and heterogeneous cohort of preterm infants represents one of the main strengths of the study. Furthermore, as a monocentric work, the enrollment, collection, and analyses of the samples took place homogeneously. On the other hand, we obtained maternal and DHA intake data through a direct interview and a specific food frequency questionnaire, thus providing only retrospectively reported data. Moreover, known placental pathologies were not considered among the exclusion criteria. Although further analyses with larger samples are needed, this study may represent a starting point to investigate new possible DHA supplementation strategies, especially in late and moderate preterm infants, even though their nutritional requirements are usually considered comparable to term babies.

## Figures and Tables

**Figure 1 nutrients-14-04595-f001:**
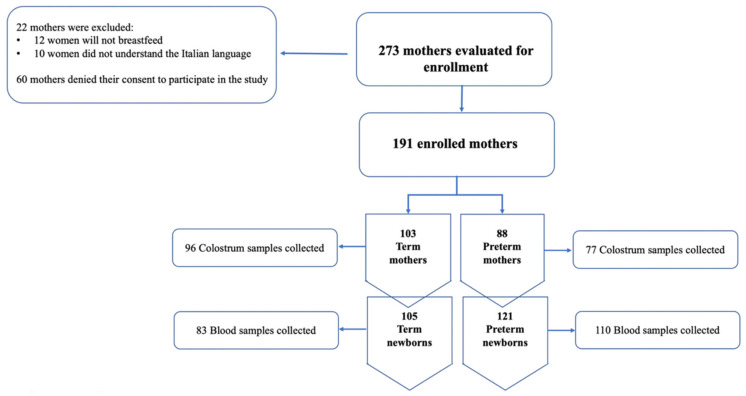
Flow-chart of the study.

**Figure 2 nutrients-14-04595-f002:**
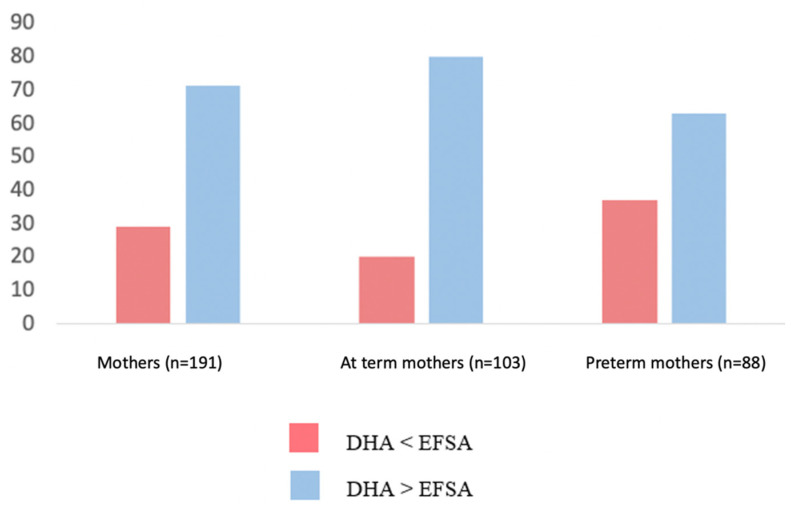
Percentage of preterm and term mothers whose DHA intake was either < lower or higher than EFSA recommendations [16].

**Figure 3 nutrients-14-04595-f003:**
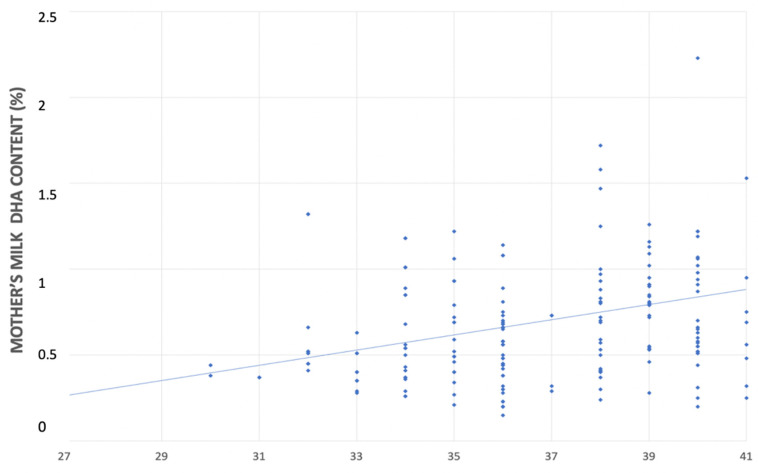
Relationship between mother’s milk DHA’s content (%) and gestational age (weeks).

**Figure 4 nutrients-14-04595-f004:**
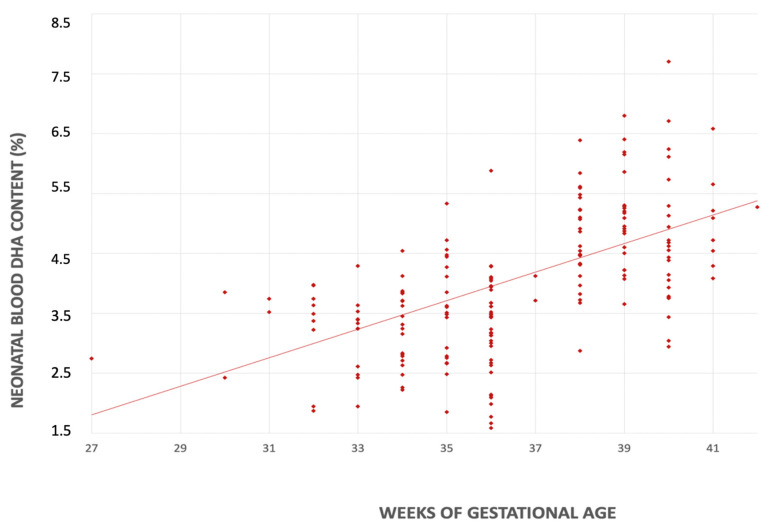
Relationship between serum DHA’s content (%) and gestational age (weeks).

**Table 1 nutrients-14-04595-t001:** Neonatal main characteristics (AGA: adequate for gestational age; SD: standard deviation).

	Newborns (n226)		Term Newborns (n105)		Preterm Newborns (n121)	
	Mean	(±SD)	Mean	(±SD)	Mean	(±SD)
**Birth weight (g)**	2743.3	730.5	3349.5	378.4	2217.3	525.5
**Gestational age (weeks)**	36.7	2.7	39.14	1.113	34.5	1.67
**Twins n(%)**	71 (31.4%)		4 (3.8%)		67 (55.3%)	
**Newborn AGA n(%)**	191 (84.5%)		93 (88.5%)		98 (81%)	

**Table 2 nutrients-14-04595-t002:** Fatty acid composition of mother’s milk. The fatty acid levels are expressed as a percentage of the total fatty acid concentration, (g DHA/100 g of total fatty acids = % FA). *p*-values: from *t*-test between preterm vs. at term population. Data are expressed as mean ± SD or median [IQR].

Fatty Acids	Population (*n* = 191)	Term Population (*n* = 103)	Preterm Population (*n* = 88)	Sign. (*p*)
	Mother’s Milk Samples (*n* = 173)	Mother’s Milk Samples (*n* = 96)	Mother’s Milk Samples (*n* = 77)	
**16:0**	26.63 (±2.01)	26.51 (±2.07)	26.78 (±1.94)	0.378
**16:1n7**	1.83 [1.51–2.20]	1.78 [1.48–2.12]	1.96 [1.62–2.38]	0.004
**18:0**	6.68 [6.04–7.45]	6.90 [6.18–7.49]	6.4 [5.85–7.28]	0.059
**18:1n9**	45.45 [43–47.53]	45.42 [43.12–47.52]	45.46 [42.72–47.53]	0.829
**18:1n7**	2.56 [2.22–3.06]	2.53 [2.25–3.04]	2.64 [2.20–3.10]	0.393
**18:2n6 (LA)**	11.81 [10.36–13.27]	11.47 [10.22–12.82]	12.31 [10.46–13.93]	0.113
**18:3n3 (ALA)**	0.38 [0.32–0.47]	0.38 [0.33–0.50]	0.37 [0.31–0.46]	0.334
**20:3n9**	0.04 [0.03–0.07]	0.04 [0.03–0.07]	0.04 [0.03–0.07]	0.943
**20:3n6**	0.82 [0.68–1.04]	0.86 [0.71–1.05]	0.76 [0.64–1.03]	0.133
**20:4n6 (ARA)**	1.01 [0.87–1.24]	1.05 [0.88–1.26]	1.0 [0.83–1.18]	0.110
**20:5n3 (EPA)**	0.08 [0.05–0.28]	0.08 [0.06–0.13]	0.11 [0.05–0.45]	0.060
**22:0**	0.19 [0.14–0.26]	0.19 [0.14–0.27]	0.2 [0.14–0.26]	0.273
**22:5n3**	0.27 [0.20–0.40]	0.33 [0.23–0.43]	0.22 [0.16–0.32]	0.276
**24:0**	0.27 [0.20–0.34]	0.29 [0.23–0.37]	0.22 [0.17–0.30]	0.006
**22:6n3 (DHA)**	0.59 [0.44–0.86]	0.71 [0.52–0.94]	0.51 [0.38–0.69]	0.001
**24:1**	0.47 [0.36–0.57]	0.49 [0.41–0.59]	0.42 [0.33–0.54]	0.010
**SFA**	33.63 [32–35.40]	33.72 [32.07–35.38]	33.59 [31.97–33.56]	0.553
**MUFA**	50.35 [48.24–52.66]	50.28 [48.26–52.84]	50.57 [48.18–52.51]	0.933
**PUFA**	15.44 [14.03–17.07]	15.23 [13.81–17.05]	15.48 [14.09–17.12]	0.234
**Omega-3 TOT**	1.49 [1.19–1.91]	1.57 [1.23–1.96]	1.4 [1.07–1.74]	0.003
**Omega-6 TOT**	13.81 [12.28–15.43]	13.57 [12.20–15.03]	14.17 [12.42–15.70]	0.178

## Data Availability

The data presented in this study are available on request from the corresponding author.

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
