# Peer review of "Comparative Analysis of Docosahexaenoic Acid (DHA) Content in Mother’s Milk of Term and Preterm Mothers"

_nutrients, 2022, doi:10.3390/nu14214595_

Round 1

Reviewer 1 Report

The study is presenting an important aspect regarding fatty acids intake in mothers and their passage through placenta to term and preterm newborns. The idea that preterm newborns need more DHA is based on fact that there are more newborns deficient in DHA in preterm babies than in those at term. Still, we do not have any information about:

1. the length of the study. were the samples collected in the same period of the year? was the study performed in a number of years? 88 preterm newborns represent a high incidence of premature births compared with 105 in the same period. Were excluded cases with placental pathologies?

2. was not discussed a possible influence of the season (in winter are more fats consumed).

3. what was the correlation between DHA in colostrum and blood of newborn? at term and preterm?

4. the determination methods of fatty acids are not mentioned. CV% intra- and interassay? for longer period of time (for example if the sample collection was made for a period of two years at minus 20C, the stability of parameters might be affected).

5. minor editing - r 209 - Other Authors

Reviewer 2 Report

Brief summary:

The authors conducted a study to compare the DHA content in colostrum of preterm mothers with that of term ones. As a secondary objective, we aimed to investigate the changes in DHA content in colostrum according to the week of gestational age.

There are important points to include or improve in the manuscript and an important lack of information, although it could be well situated within the Journal Scope, in this current presentation is not suitable to be published.

Broad comments

1. Authors could include information about preterm neonates’ population. The criteria to stablish the preterm age, their specific characteristics, risks, needs… to introduce the importance of this study.

2. Methods: This is a study focused on DHA content and there is an important lack of information about how the authors determined something so essential as fatty acid composition of maternal colostrum.

3. Results description should be improved. In most of the cases, authors indicate the information included in the figures or tables, but they do not describe the results.

4. Discussion: The authors should consider providing a more in-depth discussion of the results in the present study, and how this article addresses those results.

5. I recommend that the authors submit this manuscript to an editor for whom English is a first language. There are grammatical errors throughout the manuscript. While the large majority do not affect understanding, the article would be easier to read with corrections. Additionally, some phrasing is difficult to understand or changes the meaning such that it is incorrect as written.

Specific comments

- Line 73: authors should include the Ethics Committee approval refence number.

- Line 75: authors should include the number of women enrolled.

- Line 77: why did the authors excluded non-Italian speaker’s women?

- Line 97: authors should take into account that many studies consider that from 48-72h after delivery, mothers do not produce colostrum, thus mother milk.

Round 2

Reviewer 2 Report

Although the authors have made an effort to correct and improve their manuscript, it still lacks novelty and scientific relevance. I am aware that the study population they are working with is complicated and requires a lot of effort, but they do not provide anything that was not already known, and that is the need to supplement in DHA due to the low contents.

It is true that the quality of their article is not the motivating cause for my consideration, as the authors have done a good job correcting the article, but the relevance is low. Further follow-up of neonates could have justified the interest in this study.